# Effect and process evaluation of a multicomponent public health intervention to increase the use of primary care-based mental health services amongst children with a migrant background in Stockholm, Sweden: a protocol for a mixed-methods study

Vanessa Place  ,[1] Karima Assel,[1] Ana Hagström,[2] Ester Gubi,[1] Hanna Augustsson,[2] Christina Dalman,[1] Anna-Clara Hollander  [1]

For numbered affiliations see end of article.

**Correspondence to**
Vanessa Place;
vanessa.place@stud.ki.se

## ABSTRACT

**Introduction** The overall aim of the project is to understand how to increase access to, and use of, primary care-based mental health services for children and youth from a migrant background with mild to moderate mental health problems.

**Methods and analysis** The study will be undertaken in Haninge municipality in Stockholm, Sweden. The study has three intervention components: (1) A health communication intervention targeting parents of children/youth with a migrant background; (2) Training of professionals and volunteers who potentially have contact with parents and children with a migrant background, in order to increase the number of referrals to primary care-based mental health services, and (3) Increasing access to care at a primary care-based mental health service for children, using various strategies to lower barriers to care. The complex multicomponent intervention will be studied with an effect and a process evaluation methodology.

**Ethics and dissemination** All planned studied are approved by the Swedish Ethical Review Authority dnr 2017-135-31/5, 2019-06275, 2020-03640, 2020-06341, 2020-03642 and 2020-04180. Informed consent, written or verbal, will be obtained from all participants. The results of the project will be published continually in peer-reviewed scientific journals and disseminated to relevant stakeholders nationally and within Haninge municipality.

### Strengths and limitations of this study

► This multicomponent public health intervention includes a short-term and long-term effect and process evaluation, using a range of methods including presurveys and postsurveys, analysis of register data from multiple registers and individual and focus group interviews.

► The intervention, through both language-specific and more general components, targets both youth and parents from a migrant background, and professionals and volunteers who potentially have contact with this group.

► The study is being conducted during the COVID-19 pandemic, which has transferred the interventions online and could influence staffing levels at the primary care-based mental health services.

► The lack of available instruments to evaluate complex interventions led the research group to develop their own survey tools, in combination with previously tested questionnaires, which could reduce the reliability of findings.

► Due to its resource intensity, the project was limited to one municipality in Sweden, which may limit the generalisability of its findings to other contexts.

## INTRODUCTION

In the Scandinavian universal healthcare context, studies indicate that despite higher need of psychiatric care,[1–4] there is lower psychiatric service utilisation among all migrants,[5] including migrant children and youth, compared with their peers.[6–8] The level of psychiatric care utilisation among migrant children is particularly low for primary care-based mental health services[6] and specialist outpatient care when compared with emergency psychiatric care.[6 9] Studies have indicated that utilisation of psychiatric care is lower among migrant youth during their first years in their new country, but that levels do

increase over time.[8] The reasons behind the underutilisation of psychiatric services among children with a migrant background are not yet well understood, but a wide range of barriers to mental healthcare are likely to contribute.[10] International studies suggest that the barriers to mental healthcare among migrant children and youth are generally similar to those faced by their peers,[10] but that migrant children face additional barriers linked to their minority status.[10] Overcoming general, as well as migration-specific, barriers will be critical to meeting the mental health needs of migrant children and youth.

There are few studies on interventions to overcome barriers to mental health services for migrant children. Research has, however, indicated that schools and social agencies play an important role in referring children from migrant backgrounds to mental health services.[11] Additionally, a scoping review mapping the interventions to increase care seeking among adult migrant groups for stigmatised conditions, including mental health, concluded that health communication strategies and complex, multicomponent interventions were promising strategies[12] whereas one-component interventions (such as translating information materials) had little effect.[12]

In the absence of scientifically evaluated methods to overcome barriers to mental health services faced by migrant children, local models have developed. Since 2010, the Stockholm Region's centre for migration and health, Transkulturellt Centrum (www.transkulturelltc entrum.se/eng) has offered regular training to healthcare staff on migration, health and transcultural issues, as well as group-based health communication (henceforth called health communication) to migrants in their mother tongue. Health communication is run by individuals with a medical or public health background.

Drawing together the available evidence on barriers to care, we have designed a multicomponent complex public health intervention study that targets, in parallel: key professionals and volunteers, parents with a migrant background, and the structural barriers faced when contacting mental health services. Given the low use of primary care-based mental health services in Stockholm and the potential benefits of early interventions, the 'Bridging barriers to care for children with a migrant background' (given the acronym SAM-TINA in Swedish) research group (referred to herein as the research group) seeks to improve access to, and use of, primary care-based mental health services for children with a migrant background.

## Study aims and objectives
For the purposes of the project, migrant background is defined as children under 18 years of age who have themselves, or both of their parents, been in Sweden for less than 10 years.

The overall aim of the project is to investigate how to increase access to, and use of, care from primary care-based mental health services for children and youth with a migrant background with mild to moderate mental health problems.

## METHODS AND ANALYSIS
### Site
The study will be undertaken in Haninge municipality in Stockholm, Sweden. In 2020, Haninge had a population of 93 282, 28% of whom were born outside Sweden (compared with 19.7% in Sweden as a whole). Eleven per cent of those aged 0–17 years in Haninge in 2020 were born outside Sweden. The share of children who have themselves, or both of their parents, been in Sweden for less than ten years (our target population) will be calculated as part of the effect evaluation. However, as Swedish registers do not record race, culture, or ethnicity data (only country of origin) the proportions of different cultural backgrounds will not be calculated.

The primary care-based mental health service in focus is Psykosociala mottagningen at the primary care centre Aleris Rudan (https://www.aleris.se/har-finns-vi/stock-holm/aleris-rudans-vardcentral/) in Haninge, herein referred to as 'Rudan' and 'the primary care centre', respectively. Rudan is the second largest primary care-based mental health service, by number of visits, in Stockholm County. Despite many new patients, and the high proportion of the population in the catchment area who are foreign-born, Rudan has received very few patients with a migrant background in recent years.

### The three-component intervention
For an overview of the three-component intervention and the effect and process evaluation, see figure 1.

SAM-TINA has three components:

### Component 1: a health communication intervention targeting parents of children/youth with a migrant background
#### Component 1 in detail
The health communication intervention will be delivered in one session by health communicators, either at a venue (see 'Planning and preparations for component 1') or digitally (in line with restrictions due to the COVID-19 pandemic). The health communication sessions are offered to parents of children and youth with a migrant background and are delivered over a period of 6–12 months. Each session is approximately 2.5 hours long, with a minimum and maximum of 2–3 and eight participants, respectively. Sessions are delivered in participants' mother tongue (Arabic, Somali, Tigrinya, Amarinya or Farsi), in simple Swedish, or with an interpreter. Sessions are offered primarily in languages deemed most common in the community by staff members at venues hosting the sessions. Sessions in other languages are offered on active interest from participants.

#### Planning and preparing for component 1
In 2019–2020, the research group and Transkulturellt Centrum worked to create partnerships with units of the

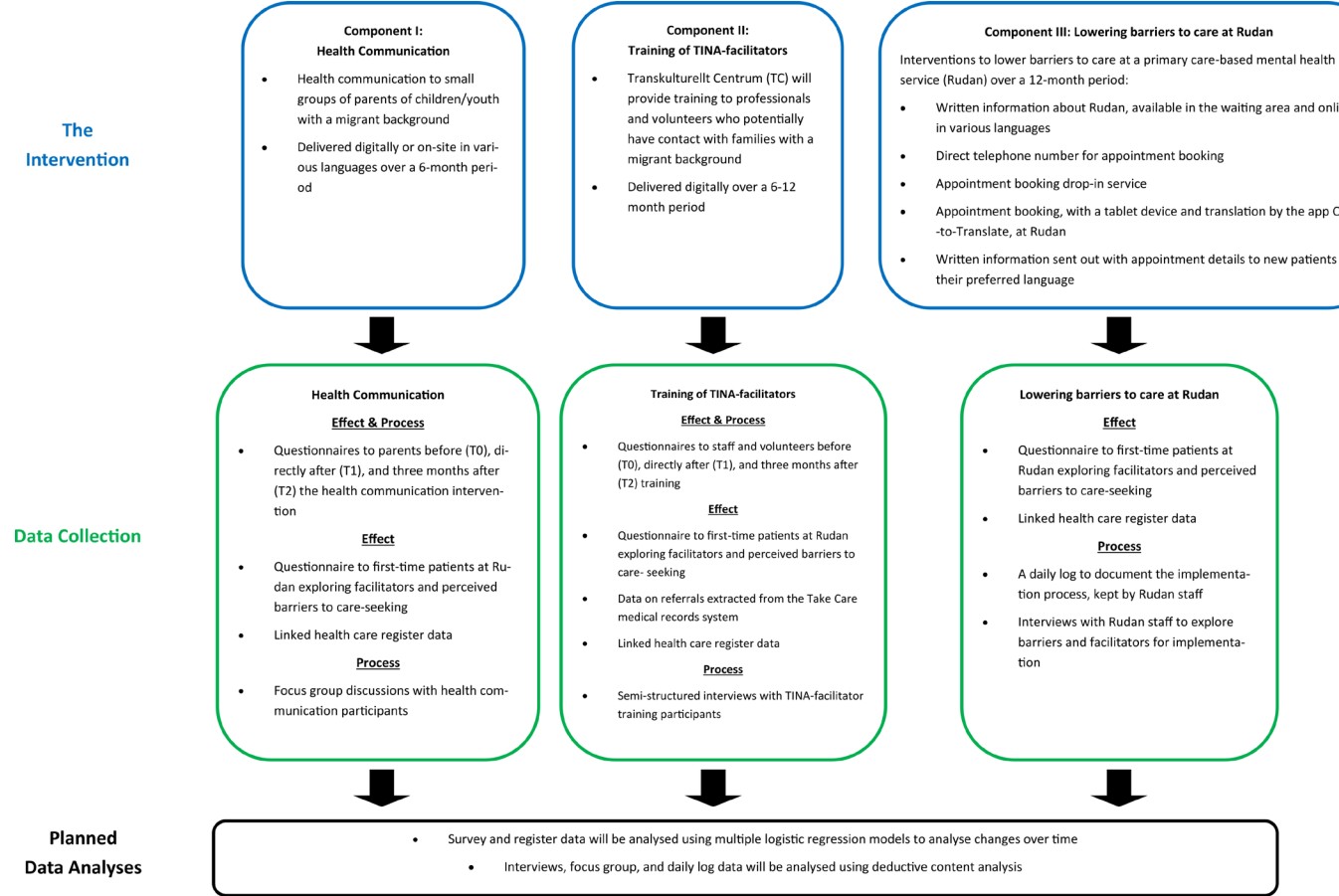

**Figure 1** A schematic overview of the three component intervention and the effect and process evaluation.

municipality and non-profit organisations in Haninge. Suitable hosts and venues for were identified with the help of the municipality's coordinators for refugee integration and parental support, 'Swedish for Immigrants' classes in Haninge, parental support groups run by the local municipality targeting newly-arrived parents, toddler-parent groups, and the Swedish Church. Recruitment, group scheduling, and later COVID-19 adjustments were planned in joint meetings between these stakeholders, Transkulturellt Centrum, and SAM-TINA researchers. Additional venues may be added during the study.

Workshops with health communicators at Transkulturellt Centrum aimed to increase their knowledge of child mental health and the organisation of mental health services. These workshops were led by an experienced child psychologist and child psychiatrist. In addition, a specific tool kit was developed, in Swedish, consisting of PowerPoint slides with discussion points and case discussions, covering six chapters:

1. Normal development compared with signs of mental distress among children and adolescents.
2. Parenting skills.
3. Potential barriers to care from a migrant perspective.
4. Available mental health services and care levels.
5. Services at Rudan and the role of a psychologist.
6. How to contact Rudan.

The content of this toolkit was informed by the expertise and experience of professionals who have contact with our target population, including health communicators themselves. Health communicators can then adapt the toolkit to the needs of each specific group (in terms of language, for example).

### Evaluation measures of component 1

► Follow-up questionnaires will be distributed before, directly after, and 3 months after the intervention by post, email, or by hand to parents who have signed up for the sessions and consented to be contacted. In the absence of a validated instrument, this questionnaire was created for SAM-TINA. All questionnaires have been translated to the language the session is delivered in.

► A questionnaire for patients visiting Rudan for the first time will be given out to all new visitors, by email or by hand. The questionnaire explores why visitors chose to seek help at Rudan and how they accessed the service. For patients younger than 14 years old, a caregiver, if present, will be asked to fill in the questionnaire.

► Analysis of register data from multiple registers, including the Psychiatry Sweden database, comparing our target population with non-migrant children

(https://ki.se/en/gph/epidemiology-of-psychiatric-conditions-substance-use-and-social-environment-epicss) as done in,[5–7] see 'The effect evaluation of the three-component intervention'.

► Focus group interviews, see 'The process evaluation of the three-component intervention'.

### Component 2: training for professionals and volunteers (herein referred to as TINA-facilitators) who potentially have contact with parents and children with a migrant background, in order to increase referrals to Rudan
*Component 2 in detail*

Training of TINA-facilitators is implemented and moderated by Transkulturellt Centrum's staff (except for the volunteer training, which is organised by the research group). The course length and format varies, depending on participating organisation's ability to allow staff to attend. All training is delivered digitally to minimise the spread of COVID-19. Most are 1-day sessions but can be two half-day, or three short, sessions. Online training for preschool teachers, with prerecorded material, is accessible for 4–6 weeks.

*Planning and preparing for component 2*

The research group mapped the demographics of Haninge municipality (see the 'Site' section), including mapping the number of children with a migrant background by residential area, in order to identify priority neighbourhoods for the intervention. Four neighbourhoods were identified. In 2019–2020, the research group and Transkulturellt Centrum worked to recruit healthcare, municipality, and community organisations in Haninge. We primarily invited units serving children in the four areas to participate in the training intervention. To tailor the content, format, and duration of the training to the diverse needs of the participating units, nine group interviews were conducted. Based on the results of these interviews, and Transkulturellt Centrum's previous experience with providing staff training, seven separate training programmes were designed. All are a minimum of 4 hours with a common core of lectures (see below), based on the interviews with professionals on their training needs and the migrant experience:

1. How to identify mental distress among children and adolescents.
2. Potential mental health consequences of migration.
3. Barriers to care for migrants in Sweden and how this can differ between groups.
4. Mental health services at Rudan.

Moreover, lectures were added to cover specific needs of the individual units, such as how to differentiate learning disabilities from post-traumatic stress for the school healthcare staff. After the training, all participants will be offered information, digitally or on paper, about Rudan (in Polish, Russian, Arabic, Turkish, Spanish, Somali, Tigrinya, English, French, Persian/Dari and Swedish) to distribute to eligible families.

*Evaluations measures of component 2*

► Questionnaires, including short-term effect and process outcomes, will be sent by email to TINA facilitators who signed up for the training and have consented to be contacted before, directly after, and 3 months after the training. These questionnaires are a combination of a modified version of a questionnaire previously used in the region for similar purposes,[13] and items on readiness for change from the implementation outcome measure.[14] They also include questions from a previously used survey[15] to measure learning and reactions to evaluate the training experience.

► A similar, shorter version of the component 2 questionnaire has been developed for the volunteer training.

► The questionnaire to first-time patients at Rudan, see 'Evaluations measures of component 1'.

► Analysis of register data, see 'Evaluations measures of component 1'.

► Semistructured interviews, see 'The process evaluation of the three-component intervention'.

### Component 3: increasing access to care at Rudan through various strategies to lowering barriers to seeking care
*Planning and preparing for component 3*

Interviews with directors and staff at primary care-based mental health services for children were conducted to explore barriers to care and support given to migrant parents (published in Swedish).[16] The research group then worked with Rudan's manager and the head of the primary healthcare centre to anchor the project, and a project coordinator was appointed at Rudan. Through discussions with the project coordinator, staff at the primary health centre, Rudan's psychologists and Transkulturellt Centrum's health communicators, the research group suggested the following interventions to lower barriers to care for migrant families:

► Written information (leaflets, posters and business cards) about Rudan available in the primary healthcare centre waiting area and on its website (https://www.aleris.se/har-finns-vi/stockholm/aleris-rudans-vard-central/psykosocial-mottagning-barn-och-ungdom/) in Polish, Russian, Arabic, Turkish, Spanish, Somali, Tigrinya, English, French, Persian/Dari and Swedish.

► A direct telephone number for appointment booking, without a phone menu, available from March 2021. The phoneline is manned from 14:00 to 15:00 hours during the clinic's regular opening hours. Messages left outside of these hours will be listened to once a day and individuals contacted.

► Appointment booking drop-in service at Rudan's reception, available from March 2021 between 14:00 and 15:00 hours during regular opening hours.

► A tablet available in Rudan's reception, with translation provided by the app Care-to-Translate (https://caretotranslate.com/sv/), for appointment booking from April 2021.

► A leaflet sent out to all new patients at Rudan, in the child's or parent/caregiver's preferred language, together with appointment details and containing information on what to expect from their first visit. Available from June 2021.

### Evaluations measures of component 3
► Questionnaire to first-time patients at Rudan, see 'Evaluations measures of component 1'.
► Analysis of register data, see 'Evaluations measures of component 1'.
► A daily log, kept by the project coordinator, to document any implementation issues.
► Interviews with Rudan's staff to explore challenges in the implementation process and to understand barriers and facilitators for implementation.

### Time frame
The three components are implemented in parallel in order to create a synergistic effect:

Component 1: Health communication sessions are currently being offered over a period of 6-12 months (planned start in November 2020 was delayed until February 2021).

Component 2: Trainings are organised over a period of 6 months (start date: November 2020, completed: May 2021).

Component 3: Interventions will be implemented over a 12-month period from March 2021.

### Recruitment and eligibility
#### Component 1: health communication interventions targeting parents of children/youth with a migrant background
*Recruitment of participants*
Recruitment of parents is currently ongoing. Information in multiple languages is being disseminated to parents by health communicators and hosts at the venues (see 'The three-component intervention'). In addition, short video clips in Arabic, Tigrinya, Somali, Farsi and easy Swedish have been developed for the project and will be disseminated online. They can be viewed at the project homepage (https://ki.se/gph/informationstraffar-for-foraldrar-i-haninge).

*Eligibility*
All parents to minors are eligible to take part, independent of country of birth, length of stay in Sweden, or language. However, a minimum of 2–3 participants are required for a session, which could lead to exclusion of parents with a less commonly spoken preferred language. All participants are welcome in the sessions, independent of literacy level, but illiterate participants are excluded from the follow-up evaluations due to the nature of the questionnaire.

#### Component 2: training of TINA-facilitators who potentially have contact with parents and children with a migrant background, to increase the number of referrals to Rudan
*Recruitment of participants*
Recruitment for the TINA facilitator training for professionals was achieved mainly with the help of key staff members, unit directors and posts on the municipality's intranet. Some of the training programmes were advertised in Transkulturellt Centrum's overall training programme which reaches healthcare staff within the Stockholm Region. The recruitment of participants for the TINA -facilitator training for volunteers engaged in civil society with Haninge was assisted by key stakeholders and contact with community organisations.

*Eligibility*
Individuals who are expected to have contact with children, adolescents and families with a migrant background are eligible to participate. This includes staff from the primary healthcare centres, paediatric clinics, adolescent health and addiction centres, social services, preschools, schools and non-profit organisations (such as sports teams). Units who serve the populations in the four prioritised neighbourhoods (see the 'Site' section) were prioritised for recruitment.

#### Component 3: increasing access to care at Rudan through various strategies to lowering barriers to seeking care at this service
*Recruitment of participants and eligibility*
All children under 18 years of age who have themselves, or both of their parents, been in Sweden for less than 10 years and who have visited the primary healthcare centre are eligible to participate.

### The effect evaluation of the three-component intervention
An effect evaluation aims to establish the causal association between an intervention in the target population and its intended outcomes.

### Primary aim
The primary aim of the effect evaluation is to assess if the three components together increase the number of children and youth with a migrant background seeking care at Rudan for mild to moderate mental health problems for the first time.

### Research questions
1. Will the observed rate of children seeking care at Rudan for the first time exceed the expected during the 12 months following the start of the intervention (exposed period), based on the rate of access in the 24 months preceding the intervention (unexposed period), and is there a difference between children and youth with and without a migrant background?
2. Will there be a change in the demographic composition of children seeking care at Rudan following start of the intervention (eg, own or parents' country/region of birth, time in Sweden, migrant status and age)?

3. Will the observed rate and demographic composition of children seeking care at Rudan for the first time during the 12 months following the start of the intervention, based on the rate in the 24 months preceding the intervention, differ as compared with other primary care-based mental health services in the Stockholm Region, and is there a difference between children and youth with and without a migrant background?

*Outcome measure*

Number of first visit consultations at Rudan (onsite and digital) before and 3, 6 and 12 months following the start of interventions.

Secondary aim

The secondary aim of the effect evaluation is to investigate if the three individual components work as they were intended to.

*Research questions*

1. Will there be a change in the self-perceived knowledge and attitudes among staff and/or volunteers exposed to the training component? (eg, increase in knowledge of signs of mental distress, change in attitudes towards working with families with a migrant background)?
2. Will the observed rate of referrals to Rudan from units exposed to the training exceed the expected during the 12 months following the start of the training component, based on the rate of access in the 24 months preceding the intervention) and differ compared with unexposed units?
3. Will there be a change in the self-perceived knowledge and attitudes among parents exposed to the health communication component (eg, increased knowledge of mental health)?
4. What are the self-reported pathways to seek the service among first-time patients at Rudan following the start of the intervention?
5. Will there be a change in perceived barriers to connect with the service, among first-time patients at Rudan following start of the component 3?

*Outcome measures*

► Staff and volunteer self-reported knowledge and attitudes at baseline, directly after, and 3 months following the training component (measured using the questionnaires for staff and for volunteers).
► Parents self-reported knowledge and attitudes at baseline, directly after, and 3 months following the health communication component (measured using the health communication questionnaire).
► First-time patients self-reported pathways to Rudan and self-perceived barriers to seek care at the service (using the first visit questionnaire).
► Number of referrals to Rudan from units exposed to the training component at baseline and 3, 6 and 12 months following start of the training (measured using data extracted from registers, see 'Primary

aim: outcome measures' and from the first-visit questionnaire).

*Methods and statistical analyses*

For the primary aim we will analyse changes over time in children seeking care at Rudan and other primary care-based services in Stockholm, before and after the start of the intervention using the linked register data (as done in references 5–7) in a time-series design, where the outcome will be explored in 1 month units, using Poisson regressions. Data from children from a non-migrant background will be used as a control. This method will also be used to analyse one of the secondary aims: referrals to Rudan. For the other secondary aims we will analyse the questionnaires using multiple logistic regressions models in both cross sectional and longitudinal designs.

**The process evaluation of the three-component intervention**

The process evaluation aims to facilitate an understanding of the effects of the intervention and provide information about under which circumstances a complex public health intervention to increase help-seeking among migrants can be implemented. The process evaluation will focus on assessing implementation fidelity (ie, adherence to the intervention, and moderating factors that may influence the degree of fidelity) using a modified version of the Conceptual Framework for Implementation Fidelity (CFIF).[17]

Data collection and participants

A mixed-methods approach will be applied, including questionnaires, semistructured interviews, focus group discussions and diaries. Interview guides will be developed based on the guiding framework, CFIF.[17] The interviews and focus group discussions will be conducted and reported following the Consolidated criteria for reporting qualitative research (COREQ) checklist.[18] The data collection for each intervention component is outlined below and in table 1.

Component 1: a health communication intervention targeting parents of children/youth with a migrant background

Questionnaires at baseline, directly after, and 3 months after the health communication session are being used to assess satisfaction, usefulness and benefits of the sessions, as well as to understand the demographics of parents who were recruited and attended the sessions.

We will perform focus group discussions with health communication participants to obtain a deeper understanding of their perceptions of these sessions. These groups will be conducted participants' preferred language (Arabic, Dari, Somali or Tigrinya), and the interview guide will be translated from Swedish to this language before back-translation to Swedish to check for accuracy. Recruitment of participants for the focus group discussions will be conducted by the key actors at the venues recruiting participants to the health communication. A purposeful sampling approach will be used

**Table 1** The general process evaluation plan including general process outcome questions, areas to measure, and data sources

| Process evaluation objective | Overarching research questions | Data source and data collection |
|---|---|---|
| Adherence | | |
| Content | Were the three intervention components implemented as planned? | Questionnaires, interviews, diary |
| Coverage | What proportion of the targeted groups participated in the intervention? | Attendance lists |
| Moderating factors | | |
| Recruitment | What barriers for maintaining involvement of individuals/groups in the intervention was identified? What recruitment procedures were used to engage professionals and parents in the intervention? | Meeting minutes, interviews, surveys |
| Participant responsiveness | Did participants find intervention content useful? How satisfied were the participants with the intervention services? How did the participants perceive the relevance of the intervention? | Questionnaires, interviews |
| Strategies to facilitate implementation | Which strategies were put in place by work units participating in the training of professionals to implement learnings from the training? | Interviews |
| Context (inner and outer) | What factors at policy, organisational and work group levels affected the implementation? Which barriers and facilitators to delivery as planned can be identified? | Interviews, meeting minutes, diary |
| Quality of delivery | How did professionals perceive the impact of the intervention for example, which elements did they find to be most helpful and unhelpful in meeting their needs? | Interviews |
| Intervention complexity | How complex is this intervention? | A group of external researchers will evaluate the intervention complexity |

Modified from Hasson Systematic evaluation of implementation fidelity of complex interventions in health and social care.[17]

to ensure variation in the sample (including participants from different countries).

### Component 2: training of TINA-facilitators who potentially have contact with parents and children with a migrant background, to increase the number of referrals to Rudan

The training of professionals will be evaluated by combining data from distributed questionnaires at baseline, directly after, and 3 months after the training, as well as interviews. Attendance lists will be used to collect information about the number of recruited participants and what proportion of target group participated in this component (coverage).

Semistructured interviews are being conducted to assess participants' satisfaction with the training, whether the training led to behaviour changes, and potential moderating factors to implementation of what was learnt from the training at the workplace. To understand contextual

and organisational differences, a purposive sampling approach is being applied to recruit informants from different work units and professional groups. During the training, participants are asked to reflect on how to transfer knowledge gained during the training to their practice, and to concretise this into a plan for how to identify and refer children with mental health problems to the correct care level. The evaluation team will follow up on whether this plan has been created and implemented by contacting an assigned 'contact person' at the participating units. Contact persons will be interviewed and, if necessary, more participants will be invited for interviews to assess the aforementioned implementation outcomes.

### Component 3: increasing access to care at Rudan by various strategies to lower barriers to seeking care

A notebook will be used by staff at Rudan to document foreseen and unforeseen events related to the changes

put in place related to the third intervention component. The notebook is a register of events and processes that take place over time. It will also be used for documenting problems that arise, and to note impressions and experiences of this process.

## Data analysis

The use of mixed methods will allow quantitative and qualitative combination of data collection sources, which can validate the findings and give a deeper understanding of the implementation process.

► Qualitative analysis.

All interviews and focus group discussions will be audiorecorded and transcribed verbatim. The transcripts will be analysed using deductive content analysis[19] based on the guiding framework CFIF.[17] Information documented in the diary is an additional qualitative data source. Reported expected and unintended events during the period of implementation will be compiled and added to the analysis.

► Statistical analysis.

For survey data, statistical analyses will be used to measure changes over time within the groups of training participants and health communication participants. Data on moderating factors and fidelity will be used to predict intervention outcomes (eg, increased knowledge as assessed in the effect evaluation). Survey answers will be dichotomised (yes/no) and logistic regression or mixed logistic regression models will be used to explore possible variation over time. As sensitivity analyses, ordinal logistic regression will be performed to test the association for ordinal outcomes as a 5-score scale. Possible heterogeneity among health centres will be taken into account using multilevel models.

## Methodological limitations

SAM-TINA is resource-intensive, so a cost–benefit analysis would have been beneficial but was not feasible. Furthermore, a lack of available instruments to evaluate complex interventions meant that the research group developed the survey tools themselves. It would have been beneficial to involve the target population in the design of the health communication intervention, but the content was instead informed by interviews with professionals who have contact with this group. Finally, the fact that the intervention included just one municipality may limit the generalisability of findings to other contexts.

## Patient and public involvement

Haninge municipality and non-profit organisations within Haninge were involved in the research from the planning stage (see 'Planning and preparing for component 1–3'). Their involvement helped us to plan and design the components but not in their evaluation; this was carried out solely by the research group (with the exception of testing the pilot versions of the questionnaires). The findings will be disseminated to relevant stakeholders within Haninge municipality and nationally.

## ETHICS AND DISSEMINATION

All planned studied are approved by the Swedish Ethical Review Authority dnr 2017-135-31/5, 2019–06275, 2020–03640, 2020–06341, 2020–03642 and 2020–04180.

Informed consent, written or verbal, will be obtained from all participants. The results of the project will be published continually in peer-reviewed scientific journals (four articles planned) and presented at conferences over a minimum period of 8 years. Beyond academic output, the findings of the SAM-TINA project will be disseminated to relevant stakeholders within Haninge municipality and nationally.

**Author affiliations**
[1]Department of Global Public Health, Karolinska Institutet, Stockholm, Sweden
[2]Medical Management Centre, Department of Learning, Informatics, Management and Ethics, Karolinska Institute, Stockholm, Sweden

**Contributors** CD and A-CH conceived the project. VP, together with A-CH, KA and AH, planned the manuscript's content and structure. VP and EG wrote the Introduction, while A-CH, KA, HA and AH wrote the Methods and analysis. VP was responsible for compiling these sections into a complete manuscript with supervision from A-CH, who critically revised the manuscript at all stages. All authors reviewed the study findings and read and approved the final version before submission.

**Funding** This work was supported by Swedish research council (Vetenskapsrådet) grant number: 2018-05763. A-CH was funded by Forte grant number: 2016-00870.

**Competing interests** None declared.

**Patient and public involvement** Patients and/or the public were involved in the design, or conduct, or reporting, or dissemination plans of this research. Refer to the Methods section for further details.

**Patient consent for publication** Not required.

**Provenance and peer review** Not commissioned; externally peer reviewed.

**ORCID iDs**
Vanessa Place http://orcid.org/0000-0002-3648-4874
Anna-Clara Hollander http://orcid.org/0000-0002-1246-5804

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
