## [Reviewer comments · BMJ Open]

ARTICLE DETAILS

TITLE (PROVISIONAL)	Effect and process evaluation of a multi-component public health intervention to increase the use of primary care-based mental health services amongst children with a migrant background in Stockholm, Sweden: a protocol for a mixed-methods study.
AUTHORS	Place, Vanessa; Assel, Karima; Hagström, Ana; Gubi, Ester; Augustsson, Hanna; Dalman, Christina; Hollander, Anna-Clara

VERSION 1 – REVIEW

REVIEWER	Anna Torrens Armstrong University of South Florida
REVIEW RETURNED	17-May-2021

GENERAL COMMENTS	I have no recommendations for edits. The study protocol is well-developed and detailed including the intervention components, preparation, implementation plan, et al. The methods for evaluation are appropriate and well-described. This is an important and timely study.
--

REVIEWER	Hui Yang Monash University, School of Primary Health Care
REVIEW RETURNED	24-May-2021

GENERAL COMMENTS	1 to consider if the three interventions are also specifically relevant with the target problem and target population, for instance the contents of training (component 1 and 2) cover cultural related barriers (such as stigma, language), and service availability (component 3) if specific patients can find same cultural and language background mental health professionals. 2 to estimate size of target population and sub-groups, such as in the study region (total population of 93,282) how many migrants who under 18-year-old and moved within 10 years, and more specifically how many of them in specific cultural groups. And, how many of those population or sub groups will seek help of mental health services (prevalence and utilization rate) 3 to consider control group (non-migrant child and adolescents) 4 to consider minimum number of participants of each cultural/language group in the recruitment stage.
--

VERSION 1 – AUTHOR RESPONSE

Reviewer: 1

Dr. Anna Torrens Armstrong, University of South Florida

I have no recommendations for edits. The study protocol is well-developed and detailed including the intervention components, preparation, implementation plan, et al. The methods for evaluation are appropriate and well-described. This is an important and timely study.

We thank the reviewer for this encouraging feedback!

Reviewer: 2

Dr. Hui Yang, Monash University

(1) To consider if the three interventions are also specifically relevant with the target problem and target population, for instance the contents of training (component 1 and 2) cover cultural related barriers (such as stigma, language), and service availability (component 3) if specific patients can find same cultural and language background mental health professionals. ‘

We thank Dr Yang for this feedback! As suggested, we have explained in more detail how the training components were tailored to our target population and their needs (see Planning and Preparing for Component 1 and 2, pages 4 and 5).

With regard to service availability (Component 3), this component of the intervention is not targeted to specific cultural groups, rather generally to all groups that experience language barriers. The changes made at Rudan as part of Component 3 have therefore been designed to benefit as many language/cultural groups as possible, so guiding specific patients to professionals from the same cultural or language background was not part of this component.

(2) To estimate size of target population and sub-groups, such as in the study region (total population of 93,282) how many migrants who under 18-year-old and moved within 10 years, and more specifically how many of them in specific cultural groups. And, how many of those population or sub groups will seek help of mental health services (prevalence and utilization rate).’

Thank you for this constructive feedback – we have now clarified that the data available on the number of people living in Haninge who were born outside of Sweden does not refer exclusively to our target group, who have lived in Sweden for less than ten years (page 3, paragraph 6). This will be calculated at a later stage in the study, as part of the effect evaluation. In addition, we have clarified that data on specific cultural groups or backgrounds is not available.

With regard to prevalence and utilisation rate of care seeking, this will be calculated as part of the register based evaluation (see ‘Evaluation measures of Component 1’, page 5).

(3) To consider control group (non-migrant child and adolescents)

We thank Dr Yang for the opportunity to clarify this important point. The analyses of register data will include analysis of data from non-migrant children as a control, this has now been clarified in the manuscript (page 5, “Evaluation measures of component 1” and on page 9, paragraph 1).

(4) To consider minimum number of participants of each cultural/language group in the recruitment stage.

The minimum number of participants for the health communication intervention was two, which is stated on page 7, paragraph 4.